# Vasoprotective Functions of High-Density Lipoproteins Relevant to Alzheimer’s Disease Are Partially Conserved in Apolipoprotein B-Depleted Plasma

**DOI:** 10.3390/ijms20030462

**Published:** 2019-01-22

**Authors:** Emily B. Button, Megan Gilmour, Harleen K. Cheema, Emma M. Martin, Andrew Agbay, Jérôme Robert, Cheryl L. Wellington

**Affiliations:** 1Department of Pathology and Laboratory Medicine, University of British Columbia, Vancouver, BC V6T 1Z3, Canada; ebbutton@mail.ubc.ca (E.B.B.); meg.gilmour@gmail.com (M.G.); harleencheema@hotmail.com (H.K.C.); emmarie@mail.ubc.ca (E.M.M.); andrew.agbay@ubc.ca (A.A.); jerome.robert@ubc.ca (J.R.); 2Djavad Mowafaghian Centre for Brain Health, Department of Pathology and Laboratory Medicine, University of British Columbia, Vancouver, BC V6T 1Z3, Canada

**Keywords:** high-density lipoprotein (HDL), cerebrovasculature, endothelial cell, amyloid beta (Aβ), Alzheimer’s disease, blood-brain barrier (BBB), inflammation, cerebral amyloid angiopathy (CAA)

## Abstract

High-density lipoproteins (HDL) are known to have vasoprotective functions in peripheral arteries and many of these functions extend to brain-derived endothelial cells. Importantly, several novel brain-relevant HDL functions have been discovered using brain endothelial cells and in 3D bioengineered human arteries. The cerebrovascular benefits of HDL in healthy humans may partly explain epidemiological evidence suggesting a protective association of circulating HDL levels against Alzheimer’s Disease (AD) risk. As several methods exist to prepare HDL from plasma, here we compared cerebrovascular functions relevant to AD using HDL isolated by density gradient ultracentrifugation relative to apoB-depleted plasma prepared by polyethylene-glycol precipitation, a common high-throughput method to evaluate HDL cholesterol efflux capacity in clinical biospecimens. We found that apoB-depleted plasma was functionally equivalent to HDL isolated by ultracentrifugation in terms of its ability to reduce vascular Aβ accumulation, suppress TNFα-induced vascular inflammation and delay Aβ fibrillization. However, only HDL isolated by ultracentrifugation was able to suppress Aβ-induced vascular inflammation, improve Aβ clearance, and induce endothelial nitric oxide production.

## 1. Introduction

An emerging area of interest for high-density lipoprotein (HDL) research is its role in neurodegenerative disease. Alzheimer’s disease (AD) is the principal form of dementia and has a tremendous social and economic burden worldwide [1]. Cardiovascular disease (CVD) risk factors including hypertension, type II diabetes mellitus (T2DM), and hypercholesterolemia augment AD risk. Furthermore, cerebrovascular dysfunction, including vascular accumulation of amyloid beta (Aβ) known as cerebral amyloid angiopathy (CAA), is found in the majority of AD cases [2,3]. Therefore, a better understanding of the vascular contributions to cognitive impairment and dementia is a priority research area [4]. The inverse association of plasma apoA-I and HDL-cholesterol (HDL-C) levels with AD risk in some [5,6], but not all [7,8,9] epidemiological studies suggests that HDL may have functional properties that protect against AD that are not necessarily captured by measurement of apoA-I or HDL-C levels per se. Mechanistically, preclinical studies show that HDL-deficient AD mice, generated by genetic deletion of apoA-I, have increased vascular Aβ deposition, known as CAA, and worse cognitive performance compared to AD mice with normal HDL levels [10], whereas AD mice overexpressing apoA-I from its native promoter show the opposite phenotype with an additional improvement in central nervous system (CNS) inflammation [11]. Furthermore, oral treatment of AD mice with an apoA-I mimetic improves memory and reduces amyloid burden [12], intravenous treatment with reconstituted human HDL or recombinant apoA-I Milano reduces soluble Aβ levels in the brains of AD mice [13,14], and either treatment reduces CNS inflammation [12,14].

Our group recently described several cerebral vasoprotective functions of HDL relevant to AD using a blood-brain barrier (BBB) cell line (hCMEC/d3) and a novel 3-dimensional (3D) perfusible cerebrovascular in vitro model composed of primary human endothelial cells (EC) and smooth muscle cells (SMC). We reported that HDL from young, healthy human donors has at least four vasoprotective functions on brain vessels that are relevant to AD, including: (i) reducing Aβ vascular accumulation and increasing transport through the vasculature; (ii) inhibiting Aβ-induced endothelial activation; (iii) delaying Aβ fibrillization and (iv) inducing nitric oxide (NO) production [15,16].

It is now of interest to assess these novel AD-relevant HDL functions in clinical biospecimens, which are typically available in smaller volumes than the ~3 mL of plasma or serum required for traditional ultracentrifugation purification of HDL. Clinical assays of cholesterol efflux capacity, the best-established HDL functional assay, use plasma treated with polyethylene glycol (PEG) to deplete non-HDL lipoproteins [17,18,19]. While apoB-depleted plasma is not “HDL” as many other plasma components remain in the preparation, it is a more cost-effective and higher-throughput method than ultracentrifugation and requires much less starting material. Consequently, the benefits and weaknesses of apoB-depleted plasma must be weighed against those of HDL isolated by ultracentrifugation when developing novel assays of HDL function. Therefore, this study was designed to evaluate how the pre-analytic factors of ultracentrifugation and PEG precipitation affect assay performance of cerebrovascular functions of HDL that are relevant to AD, namely increasing Aβ vascular clearance, anti-inflammatory activity against Aβ, delay of Aβ fibrillization and promotion of NO secretion.

## 2. Results

### 2.1. ApoB-Depleted Plasma Reduces Aβ Accumulation in Bioengineered Vessels Similar to High-Density Lipoprotein (HDL) Isolated by Ultracentrifugation But Does Not Effectively Increase Aβ Transport across the Vessel Wall

We developed an in vitro, 3D model of CAA using primary human EC and SMC cells cultured under native-like flow conditions (Appendix A) and showed that when HDL isolated by ultracentrifugation is circulated through the lumen of the arteries, the accumulation of Aβ42 in the arterial wall is prevented [15]. In the present study, we used this 3D human CAA model to assess whether apoB-depleted plasma produced by PEG-precipitation reproduces the protective effects of purified HDL related to Aβ42 deposition and transport. After normalizing to cholesterol concentrations, HDL and apoB-depleted plasma were circulated at 10.5 mg/dL through bioengineered arteries. This dose mirrors that previously used [15,16] and corresponds to 20% and 25% of normal plasma HDL-C concentrations for healthy adult females and males, respectively [20]. We observed that HDL isolated by ultracentrifugation and apoB-depleted plasma were equally effective in reducing soluble Aβ42 levels in vascular tissue after 24 h; from 61.17 ng/mg to 17.49 ng/mg (*p* = 0.009) and 27.69 ng/mg (*p* = 0.036) for HDL and apoB-depleted plasma, respectively (Figure 1a), albeit with higher variability observed with apoB-depleted plasma. We also measured Aβ42 transported through the vessel wall into the circulation over the first 4 h of treatment and also at 24 h. Over the first 4 h (Figure 1b) and after 24 h (Figure 1c), only HDL significantly promoted Aβ42 transport into the circulation compared to Aβ42 alone (*p* = 0.020, *p* = 0.005 respectively). More specifically, HDL treatment increased the amount of Aβ42 transported into the circulation from 21.75 ng/mL to 25.67 ng/mL whereas Aβ42 in the circulating media after treatment with apoB-depleted plasma only reached 22.15 ng/mL. Aβ42 transport was also significantly different between tissues treated with HDL and those treated with apoB-depleted plasma over the 4 h and after 24 h (*p* = 0.0006, *p* = 0.010 respectively). Importantly, Aβ42 levels in ultracentrifuge-isolated HDL and apoB-depleted plasma were below the detection limit as assessed by enzyme-linked immunosorbent assay (ELISA), therefore, the differences in Aβ42 measured in the circulation media cannot be attributed to Aβ42 bound to HDL. However, it is possible that non-HDL plasma components, such as albumin or immunoglobulins, or residual PEG solution present in the apoB-depleted plasma, could mask the Aβ42 epitope used by the ELISA. Therefore, the lack of observed effect of apoB-depleted plasma on Aβ42 transport through the vessel wall may be due to a technical limitation.

### 2.2. ApoB-Depleted Plasma Does Not Retain the Ability of HDL to Reduce Aβ-Induced Monocyte Binding to the Endothelium

AD is associated with cerebrovascular inflammation and Aβ has been reported by several independent groups to activate EC [21,22,23]. Previously, we demonstrated that HDL isolated by ultracentrifugation from the plasma of young healthy donors reduced Aβ-induced monocyte binding to EC [16]. To asses this anti-inflammatory function of HDL in apoB-depleted plasma, we performed monocyte-binding assays where HDL or apoB-depleted plasma were circulated through 3D bioengineered arteries that were treated abluminally with Aβ42 for 21 h after which fluorescently labelled monocytes were added to the circulation for 3 h. Quantification of adhered monocytes to the vessel lumen demonstrated that circulating HDL reduced endothelial activation from a mean of 16.57 adhered monocytes to 5.68 cells but that circulation of apoB-depleted plasma did not significantly affect monocyte adhesion with a mean of 12.64 adhered cells observed (*p* = 0.011 HDL vs. vehicle, *p* = 0.435 apoB-depleted plasma vs. vehicle, *p* = 0.101 HDL vs. apoB-depleted plasma) (Figure 2a). Recently we reported that HDL suppresses Aβ-induced peripheral blood mononuclear cell (PBMC) binding to brain-derived EC in a mechanism distinct from TNFα-induced PBMC adhesion [16]. To further investigate whether the lack of response observed with apoB-depleted plasma is unique to Aβ or extends to all inflammatory stimuli, we measured PBMC adhesion to the well-characterized BBB cell model hCMEC/d3 cultured in regular tissue culture plates. First, we confirmed that apoB depleted plasma did not suppress Aβ42-induced endothelial activation in this 2D model. We observed that pretreatment of hCMEC/d3 for 2 h with HDL isolated by ultracentrifugation reduced the mean number of adhered PBMC to Aβ42 stimulated endothelial cells from 8.6 to 3.04 PBMC while pretreatment with apoB-depleted plasma did not significantly change PBMC adhesion (*p* = 0.037 vehicle vs. Aβ42 alone, *p* = 0.006 Aβ42 alone vs. Aβ42 with HDL, *p* = 0.794 Aβ42 alone vs. Aβ42 with apoB-depleted plasma, *p* = 0.050 Aβ42 with HDL vs. Aβ42 with apoB-depleted plasma) (Figure 2b). On the other hand, both preparations significantly suppressed PBMC adhesion in hCMEC/d3 stimulated with the classical inflammatory stimulus tumor necrosis factor alpha (TNFα) from a mean of 14.39 to 6.65 and 7.40 adhered PBMC for HDL and apoB-depleted plasma, respectively (*p* = 0.001 vehicle vs. TNFα, *p* = 0.007 TNFα alone vs. TNFα with HDL, and *p* = 0.017 TNFα alone vs. TNFα with apoB-depleted plasma) (Figure 2c).

### 2.3. ApoB-Depleted Plasma Is Functionally Equivalent to HDL with Respect to Delaying Aβ42 Fibrillization

We have previously shown that HDL isolated by ultracentrifugation from healthy human donors can delay Aβ42 fibrillization in a cell-free Thioflavin T assay [16]. Therefore, we used this assay to compare HDL isolated by ultracentrifugation against apoB-depleted plasma produced by PEG-precipitation and found that both preparations delayed the onset of Aβ42 fibrillization to an equivalent extent (Figure 3a). Fluorescence curves were fitted to a Boltzman sigmoidal curve to determine the time to half-maximal fluorescence (t_50_) and maximal fluorescence. The t_50_ was significantly delayed with the addition of either HDL or apoB-depleted plasma from 3.85 h to 9.22 h and 7.84 h, respectively (*p* = 0.001, *p* = 0.004) (Figure 3b). However, we note that the curve-fitting approach used here may require future optimization due to the distinct shape of the apoB-depleted plasma curve. Although the data for apoB-depleted plasma with Aβ42 fit the Boltzman sigmoidal curve well (*R*^2^ = 0.94), the sharp increase in fibrillization observed with vehicle and HDL is not evident with apoB-depleted plasma, and the maximal fluorescence observed in samples containing HDL isolated by ultracentrifugation is significantly higher compared to Aβ42 alone and apoB-depleted plasma (50,895 relative fluorescence units (RFU) vs. 30,090 RFU and 29,324 RFU respectively, *p* = 0.044 and *p* = 0.038) (Figure 3c).

### 2.4. ApoB-Depleted Plasma Does Not Induce Nitric Oxide (NO) in Human Brain-Derived Endothelial Cells

A key vascular protective function of HDL is the ability to induce NO production in EC via the phosphorylation of endothelial nitric oxide synthase [24]. We have previously shown that HDL isolated by ultracentrifugation maintains this function in brain-microvascular EC and that the mechanism by which HDL induces NO production in these cells is distinct from other anti-inflammatory mechanisms [16]. Here we compared the ability of HDL isolated by ultracentrifugation and apoB-depleted plasma to induce NO production in hCMEC/d3. Treatment with HDL for 4 h increased intracellular NO levels from a log fluorescence intensity of 2.14 RFU to 2.57 RFU (*p* = 0.028), however, the intracellular levels of NO in cells treated with apoB-depleted plasma were unchanged (*p* = 0.994 vehicle vs. apoB-depleted plasma, *p* = 0.027 HDL vs. apoB-depleted plasma) (Figure 4).

### 2.5. Non-HDL Plasma Components in ApoB-Depleted Plasma Interfere with Some Anti-Inflammatory Activities of Purified HDL

Two hypotheses were tested to explain why apoB-depleted plasma fails to reduce Aβ-induced monocyte binding and NO production. First, residual PEG polymer can remain in apoB-depleted plasma even after extensive dialysis, which may interfere with specific HDL functions due to a reduction in the hydration shell surrounding the HDL particles [25]; second, apoB-depleted plasma contains abundant non-HDL plasma proteins, which may also interfere with certain HDL functions. Two additional specimen processing protocols were used to test these hypotheses. First, we tested whether PEG interferes with HDL functions by first using PEG-precipitation to deplete apoB-containing lipoproteins followed by ultracentrifugation to remove non-HDL plasma proteins, thereby producing HDL that has been exposed to PEG polymers (Appendix A). To test whether non-HDL plasma components interfere with specific HDL activities we used a single ultracentrifugation step to produce an isolate with a similar composition to PEG-precipitated plasma but with no exposure to PEG (Appendix A).

HDL isolated by ultracentrifugation and HDL isolated by the combination of PEG-precipitation and ultracentrifugation suppressed Aβ-induced PBMC adhesion from 5.48 average cell counts to 1.93 (*p* = 0.003) and 2.89 average cell counts (*p* = 0.037), respectively, whereas neither preparation of apoB-depleted plasma had any significant effect on PBMC adhesion (Figure 5a). Similarly, HDL isolated by ultracentrifugation and by the combination method increased NO-production from a log fluorescence intensity of 3.64 RFU to 4.92 RFU (*p* = 0.047) and 4.15 RFU (*p* = 0.006), respectively, while neither preparation of apoB-depleted plasma significantly altered NO production (Figure 5b). These observations support the hypothesis that non-HDL plasma components present in apoB-depleted plasma, rather than exposure of HDL particles to PEG, are responsible for the functional differences in the brain-relevant HDL activities evaluated in this study.

## 3. Discussion

It is well-known that HDL composition varies according to the method by which this heterogeneous lipoprotein subclass is isolated [25,26,27]. It is also increasingly clear that HDL function may be more relevant than HDL-C levels when predicting coronary artery disease risk [17,18,19]. Although Mendelian randomization studies showed that high HDL-C levels do not causally contribute to reduced coronary artery disease risk [28], HDL may still have cerebrovascular functions that lower AD risk. Aging, T2DM, hypertension, and hypercholesterolemia are established AD risk factors and all can be associated with reduced HDL function [29]. Changes in HDL functions such as attenuation of inflammation in vascular diseases may partly account for the association between these diseases and AD risk. Supporting this concept is the observation that HDL isolated from AD patients displays reduced cholesterol efflux capacity and a reduced ability to suppress TNFα-induced intercellular adhesion molecule 1 (ICAM-1) expression in EC compared to HDL isolated from cognitively healthy individuals [30]. Whether HDL has additional functions that are more specific to cerebrovascular health in AD is now an important question.

The cerebrovascular system has an important role in clearing Aβ from the brain across the BBB or through perivascular clearance along the artery wall [31]. Dysfunction in the vascular clearance of Aβ is common in AD and is thought to contribute to the deposition of Aβ within cerebral arteries known as CAA [32]. Modeling Aβ clearance and CAA formation with 3D bioengineered arteries allowed us to discover that HDL circulated through the vessel lumen suppresses Aβ accumulation in the vessel wall and tends to increase Aβ recovery in the luminal circulating media [15]. Furthermore, it has been suggested that the exposure of vascular cells to Aβ during clearance from the brain may result in vascular inflammation [21,22,23]. We confirmed that Aβ42 has inflammatory effects in 3D bioengineered arteries and 2D EC cultures and further found that HDL suppressed this Aβ-induced EC activation [16]. Importantly, it is increasingly recognized that in vitro experimental data obtained using traditional 2D cell culture methods may be of limited translational relevance [33]. As our 3D bioengineered vascular model maintains the anatomical and physiological complexity of a native vessel, it may be a better model to investigate HDL function under more physiological conditions. However, as our 3D models requires substantial sample volume, this study was designed to optimize preparation method to prepare for studies using clinical biospecimens from patients with known AD risk factors such as age and cardiometabolic diseases.

Density gradient ultracentrifugation is the method by which lipoprotein subclasses were first discovered in human plasma, and remains a key method in their isolation today including use in several clinical studies [24,34]. However, this method has several established drawbacks including loss of some HDL proteins, such as apoE and apoA-I, due to high shear forces [26] and the potential redistribution of apolipoproteins due to high ionic strength during isolation and the subsequent desalting process [35]. Indeed, we observed that apoE levels were significantly lower in HDL isolated by ultracentrifugation compared to plasma and apoB-depleted plasma when normalized to HDL-C content (Appendix A). Ultracentrifugation also requires a substantial volume of starting material, is relatively costly and is low throughput. A higher-throughput method of HDL isolation uses the neutral polymer PEG to precipitate apoB-containing, non-HDL lipoproteins from plasma. ApoB-depletion using PEG is attractive for clinical studies as it is cost effective, high throughput and requires minimal specimen volume. ApoB-depleted plasma has been used previously in clinical studies to evaluate cholesterol efflux capacity and demonstrates a similar maximal efflux capacity compared to HDL isolated by ultracentrifugation [36], which we confirmed here (Appendix A). Nevertheless, it is important to note that apoB-depleted plasma is a heterogeneous preparation that retains proteins such as albumin and globulins as well as other factors that may affect certain functional assays. Furthermore, Davidson et. al. recently evaluated the use of PEG-precipitation in comparison to other precipitation reagents and found that while PEG-precipitation does not drastically alter the phospholipid or cholesterol content of particles isolated in the HDL size range compared to its profile in native serum, a finding that we also confirmed (Appendix A), the distribution of apoA-I and apoE and the cholesterol efflux capacity of specific HDL fractions was altered. Therefore, caution was suggested in using PEG-precipitation in HDL functional assays [25].

Here we show that sequential ultracentrifugation is superior to PEG-precipitation as a method to isolate HDL particles for assays of cerebrovascular functions relevant to AD. Where sample volume may be too small for ultracentrifugation, apoB-depleted plasma functioned equally as well as ultracentrifuge-isolated HDL with respect to CAA. As CAA cannot yet be evaluated in live humans, assaying HDL in 3D bioengineered arteries may provide an unprecedented proxy of human CAA risk in vivo. The ability to use apoB-depleted plasma instead of HDL isolated by ultracentrifugation may streamline efforts to evaluate the anti-CAA effects of HDL in clinical specimens. However, apoB-depleted plasma is not suitable for other AD-relevant assays, including promoting Aβ transport through the wall of 3D bioengineered arteries or NO production. Several potential mechanisms may explain the discrepancy in the effects against accumulation in the vessel wall compared to clearance into the lumen. For example, apoB-depleted plasma may prevent the entrance of abluminal Aβ42 into the vessel wall while HDL promotes the removal of Aβ42 from the vessel wall and clearance into the lumen. It is also possible that the presence of extensive plasma proteins in the apoB-depleted plasma circulating in the luminal media may mask the Aβ42 epitope and reduce its detection in luminal media by ELISA. Finally, we found that apoB-depleted plasma did not recapitulate the ability of HDL isolated by ultracentrifugation to suppress Aβ-induced vascular inflammation in 2D and 3D models or to induce NO production in brain-derived EC. Nevertheless, we demonstrated here that TNFα-induced EC inflammation is reduced by apoB-depleted plasma similarly to HDL isolated by ultracentrifugation, which is of interest for investigators studying cardiovascular disease.

The discrepancy between ultracentrifuge-isolated HDL and apoB-depleted plasma in these specific functions cannot be attributed to residual PEG in the latter preparation. This is because combining PEG precipitation with a single ultracentrifugation step produces a preparation with similar functions to the traditional ultracentrifugation method. Instead, the functional differences in PEG-precipitated plasma may result from remaining non-HDL plasma proteins that potentially interfere with certain HDL functions. That PEG-precipitated plasma and plasma subjected to a single ultracentrifugation to remove apoB-containing lipoproteins perform similarly in the assays tested here supports the interference hypothesis. It is also possible that the functional differences observed between the two plasma preparations could be explained by a loss or redistribution of apolipoproteins resulting from the high shear and ionic forces used during KBr density gradient ultracentrifugation [35,37]. In future, the use of D_2_O/sucrose as a density agent instead of KBr, as investigated by Ståhlman et al. [35], could help to investigate the effect of ionic strength on the functions of isolated HDL.

There are several limitations to this study. First, the donors who participated in this study were young and healthy, and we anticipate that at least some of the AD-relevant functions assessed here may be altered by aging or comorbid conditions associated with increased AD risk, which are associated with increased blood inflammatory markers [38]; second, a relatively small number of functional tests assayed in this study were selected for their potential relevance to AD or cerebrovascular health. Furthermore, HDL was only assessed at a single concentration, which corresponds to the most frequent concentration used in studies evaluating HDL functions in vitro [24,30,34,39,40,41], therefore presenting the possibility of effect saturations that may mask subtle functional differences by isolation method. Similarly, we evaluated only two common HDL isolation methods. Other HDL isolation methods such as immunoaffinity chromatography or fast protein liquid chromatography were not pursued here as these methods do not provide sufficient HDL for the cell-based functional assays in this study. Another limitation is that we chose to normalize preparations to HDL-C concentrations. HDL and apoB-depleted plasma preparations could alternatively have been normalized to apoA-I concentrations, however, this was avoided due to potential loss of apoA-I from HDL particles during ultracentrifugation [26]. HDL particle number is another method of normalization, which could be used in the future. Although normalization to the initial plasma volume could have been used, potential variances in product loss between isolation methods was a concern. Finally, future mass-spectrometry studies will be needed to define the compositional differences between HDL preparations and understand their association to each of the AD-relevant HDL functions tested here.

In summary, we report here that apoB-depleted plasma can be used to evaluate select HDL functions relevant to cerebrovascular health, but that ultracentrifugation is the preferred method to isolate HDL that exhibits at least 4 distinct AD-relevant vasoprotective functions. First, HDL delays Aβ fibrillization in a cell-free assay; second, HDL induces NO production in a process that is inhibited by co-treatment with large amounts of non-HDL plasma components; third, HDL inhibits Aβ-induced endothelial activation in brain-derived EC; fourth, HDL may protect against AD by preventing Aβ accumulation within cerebral vessels.

## 4. Materials and Methods

### 4.1. Blood Collection

All experiments were conducted under an approved protocol (UBC Clinical Ethics Research Board H14-03357, 10 Novermber 2015). Non-fasting blood was obtained from healthy, normolipidemic male or female volunteers aged 20 to 35 years in dipotassium ethylenediaminetetraacetic acid (K_2_-EDTA) coated plasma vacutainer tubes (BD Biosciences, San Jose, CA, USA) by venipuncture after receipt of informed consent. Plasma was isolated from whole blood by centrifugation at 1500× *g* for 10 min. Additional plasma was obtained from the Network Centre for Applied Development (netCAD) through the Canadian Blood Services under an approved protocol (Canadian Blood Services REB 2016.015, nc0027, 25 October 2017). Plasma was either immediately processed to isolate lipoproteins or aliquoted and stored at −20 °C until use. Whole blood for PBMC isolation was collected from healthy volunteers as described above and PBMC were by isolated by centrifugation on a continuous density gradient (Lymphoprep, StemCell, Vancouver, BC, Canada) following the manufacturer’s directions.

### 4.2. HDL Isolation

Except where indicated otherwise, HDL was isolated from plasma by 2 step sequential potassium bromide (KBr) gradient density ultracentrifugation [16] (Appendix A) and apoB-depleted plasma was produced by polyethylene glycol (PEG) precipitation [25] (Appendix A). HDL and apoB-depleted plasma were dialyzed in dialysis buffer (150 mmol/L NaCl, 0.3 mmol/L EDTA, pH 7.4) for 2 days at 4 °C, with the buffer refreshed 4 times.

For selected experiments, two additional isolation methods were used. First, PEG-precipitation was used to remove apoB-containing lipoproteins after which the density of the supernatant was adjusted to 1.21 g/mL and subjected to a single ultracentrifugation step to remove other plasma proteins from the HDL fraction. The upper fraction containing HDL was collected and dialyzed as above (Appendix A); second, a single step ultracentrifugation was performed by adjusting plasma density to 1.063 g/mL with KBr and centrifuging at 160,000× *g* for 16 h at 14 °C. The upper fraction containing non-HDL lipoproteins was removed and the bottom fraction containing HDL and other plasma proteins was collected and dialyzed as above (Appendix A).

### 4.3. Enzyme-Linked Immunosorbent Assay (ELISA)

ApoA-I levels were quantified by ELISA (Mabtech, Cincinnati, OH, USA, 3710-1HP-2) following the manufacturer’s directions with a 50,000-fold dilution in the supplied dilution buffer. ApoE levels were quantified using an in-house sandwich ELISA [42]. ELISA data for apoA-I and apoE were normalized to HDL-C levels for both HDL or apoB-depleted plasma preparations.

### 4.4. Cell Culture

Human cerebral microvascular endothelial cells (hCMEC/d3) (Fisher, Hampton, NH, USA) passage 30–40 were maintained in complete endothelium growth medium 2 (EGM2) (Lonza, Visp Switzerland) with 2% foetal bovine serum (FBS) (Invitrogen, Carlsbad, CA, USA) in a humidified incubator at 37 °C at 5% carbon dioxide (CO_2_). Murine RAW 264.7 macrophages (ATCC, Manassas, VA, USA) were maintained in Dulbecco’s modified eagle medium (DMEM) (Invitrogen) containing 10% heat-inactivated FBS (Invitrogen) in a humidified incubator at 37 °C at 5% CO_2_.

### 4.5. Fabrication of 3D Bioengineered Vessels

3D bioengineered vessels composed of human cells were grown on a scaffold and maintained in culture under dynamic pulsatile flow as described [15]. Briefly, 10 × 2 mm tubular scaffolds were formed from non-woven polyglycolic-acid meshes coated with 1.75% polycaprolactone (80 kDa, Millipore Sigma, Burlington, MA, USA) in tetrahydrofuran (Millipore Sigma). Human umbilical smooth muscle cells (SMC) and endothelial cells (HUVEC) were isolated as described [43] under UBC ethics protocol H14-03372; 2 × 10^6^ SMC/cm^2^ were seeded within sterilized and equilibrated scaffolds followed by 3 days in static culture and 7 days of culture under dynamic pulsatile flow in the bioreactor. 1 × 10^6^ HUVEC/cm^2^ were then seeded within the scaffolds followed by culture in static conditions for 5 days then 10 days under dynamic pulsatile flow in the bioreactor. Vessels were characterized histologically as described [15]. Briefly, O.C.T. (ThermoFisher, Waltham, MA, USA) embedded vessels were sectioned at 20 μm on a cryostat (Leica, Germany) then rehydrated with phosphate buffered saline (PBS), fixed for 20 min in 4% paraformaldehyde (PFA), washed in Tris-HCl (0.5 mol/L, pH 7.6), and blocked in 5% goat serum, 1% BSA in PBS. Immunofluorescence was performed to visualize HUVEC and SMC by incubating in primary antibodies for CD31 (Biolegend, San Diego, CA, USA, WM59, 1:50) or smooth muscle actin (Millipore Sigma, 1A4, 1:200) overnight followed by washing and incubation in secondary anti-rabbit or anti-mouse Alexa-488 or Alexa-594 antibodies (Invitrogen) for 45 min at room temperature. Sections were washed with PBS then mounted in Prolong antifade containing DAPI (ThermoFisher) and imaged on an inverted fluorescent microscope (Zeiss, Oberkochen, Germany).

### 4.6. Nitric Oxide Production

hCMEC/d3 were seeded at 5 x 10^5^ cells/well in 6-well plates (Fisher, Hampton, NH, USA) and cultured in complete EGM2 for 3 days before serum starvation (0.2% FBS) for 16 h. Cells were treated with 50 μg/mL HDL isolated by ultracentrifugation or an equivalent amount of apoB-depleted plasma based on HDL-C concentration, both from the same plasma donor, along with 1 μmol/L of 4,5-diaminofluorescein diacetate (DAF2-DA, Cayman Chemical, Ann Arbor, MI, USA) in EGM2 lacking vascular endothelial growth factor (VEGF) and 0.2% FBS. After 4 h, cells were washed with PBS and trypsinized for 5 min. Triazolofluorescein fluorescence was measured (excitation 485 nm, emission 538 nm) using an Infinite M200Pro plate reader (Tecan, Männedorf, Switzerland).

### 4.7. Preparation of Aβ Monomers

Recombinant Aβ42 peptides (California Peptide, Napa, CA, USA) were dissolved in ice-cold hexafluoroisopropanol (HFIP, Millipore Sigma) and aliquoted. HFIP was removed by evaporation overnight and the Aβ film was stored at −20 °C until use. The film was reconstituted in DMSO to 5 mmol/L followed by a further dilution to 100 μmol/L in FBS free EBM2 containing 0.2% BSA immediately before use.

### 4.8. Anti-Inflammatory Function

hCMEC/d3 were seeded at 1 × 10^5^ cells/well in 24-well plates (Fisher, Hampton, NH, USA) for 3 days until confluent. Cells were primed for 2 h with 50 μg/mL HDL isolated by ultracentrifugation or an equivalent amount of apoB-depleted plasma based on cholesterol concentration, both from the same plasma donor, in FBS-free EGM2 containing 0.2% BSA then treated with 1 ng/mL TNFα (Preprotech, Rocky Hill, NJ, USA) or 0.1 μmol/L Aβ monomers for 3 h. The dose of Aβ used here was selected based on our previous work investigating the response of PBMC adhesion to increasing doses of Aβ [16]. PBMC previously labeled for 30 min with Cell-Tracker Red (Invitrogen) following the manufacturer’s instructions, were added at a density of 5 × 10^5^ cells/well and allowed to adhere for 3 h. Notably, as the media containing the treatments was not removed between steps of this assay, the final conditioned media contained HDL or apoB-depleted plasma in combination with Aβ42 or TNFα and labelled PBMC. Cells were extensively washed with PBS, fixed with 4% PFA for 15 min, washed with PBS and counterstained with 4,6-diamidino-2-phenylindole (DAPI, Sigma Millipore) (100 ng/mL). Adhered PBMC were imaged using an inverted fluorescent microscope (Zeiss) and counted in 5 random squares of 7.84 mm^2^ using ImageJ (National Institutes of Health, Bethesda, MD, USA).

### 4.9. Cell-Free Assay of Aβ Fibrillization

The formation of Aβ fibrils was measured using Thioflavin T, an indicator of β-sheet formation, in a cell-free assay. Infinite M2000 Pro plate reader (Tecan). 10 μmol/L monomeric Aβ42 were incubated in a buffer consisting of Thioflavin-T (20 mmol/L in 150 mmol/L NaCl and 5 μmol/L of HEPES at pH 7.4), with and without 0.1 mg/mL HDL isolated by ultracentrifugation or an equivalent amount of apoB-depleted plasma based on HDL-C concentration, both from the same donor in a 96-well, black plate. The plate was kept at 37 °C and subjected to orbital shaking for 20 s every 5 min using an Infinite M2000 Pro plate reader. Fluorescence intensity was measured every 5 min for 12 h in total (440/490 nm excitation/emission).

### 4.10. Amyloid Beta Clearance and Accumulation Assays Using Engineered Vessels

Aβ42 monomers were injected within the tissue chamber (the “brain side”) to a final concentration of 1 μmol/L, a dose chosen based on a previous dose-response experiment [15]. Immediately afterward, the bioreactor media was substituted with complete EGM2 media (2% FBS) containing 200 μg/mL HDL isolated by ultracentrifugation or an equivalent amount of apoB-depleted plasma based on cholesterol concentration, both from the same plasma donor. Luminal medium was collected from the circulation chamber (the “blood side”) from 5 min to 4 h and 24 h after treatment. At 24 h bioengineered tissues were collected and lysed in RIPA buffer (10 mM Tris pH 7.4, 150 mM NaCl, 1.0% NP-40, 1.0% sodium deoxycholate, 0.1% SDS and cOmplete protease inhibitor with EDTA (Roche, Basel, Switzerland)). Aβ42 (KHB3482, ThermoFisher) was quantified in circulating media and tissue lysates using commercial ELISA and tissue concentrations were normalized to total protein content measured by BCA protein assay kit (ThermoFisher).

### 4.11. Amyloid Beta Induced Monocyte Adhesion Using Engineered Vessels

As above, Aβ42 monomers were injected within the tissue chamber (the “brain side”) to a final concentration of 1 μmol/L, a dose selected from a previous dose-response experiment [16]. Immediately afterward, the bioreactor media was substituted with complete EGM2 media (2% FBS) containing 200 μg/mL HDL isolated by ultracentrifugation or an equivalent amount of apoB-depleted plasma based on cholesterol concentration, both from the same plasma donor. After 21 h, Cell-Tracker Red labelled THP-1 monocytes were added to the circulating media (1 million cells/mL) and allowed to adhere for 3 h. Bioengineered tissues were then collected, fixed in 4% PFA in PBS, rinsed in PBS, and the entire tissue thickness was imaged using a SP8 confocal microscope (Leica). Adhered monocytes were counted in 3 random squares of 1.23 mm^2^.

### 4.12. Characterization of HDL Preparations

The protein concentration of HDL was quantified using the Pierce BCA Protein Assay Kit (ThermoFisher, Waltham, MA, USA) following the manufacturer’s directions. HDL cholesterol (HDL-C), phospholipid, free and esterified cholesterol, and triglycerides were quantified using commercially available kits (Wako Diagnostics, Mountain View, CA, USA) following the manufacturer’s directions.

### 4.13. Gel Electrophoresis

HDL, apoB-depleted plasma, and total plasma were analysed using native and sodium dodecyl sulphate (SDS) polyacrylamide gel electrophoresis. HDL (10 μg of protein) and an equivalent amount of apoB-depleted plasma based on HDL-C concentration, both from the same plasma donor, were separated by electrophoresis, using 6% acrylamide for native and 10% for denaturing, respectively. To visualize all proteins, gels were washed in water, stained with Coomassie R-250 (0.1% Coomassie R-250, 40% methanol, 10% acetic acid) overnight, destained in a solution of 10% methanol and 7.5% acetic acid in water for at least 2 h, and imaged using a ChemiDoc MP imager (Bio-Rad, Hercules, CA, USA).

### 4.14. Statistical Analysis

Statistical analyses were performed with GraphPad Prism 6.05 software (GraphPad, San Diego, CA, USA) and *p* < 0.05 was considered significant. Data were obtained from at least three independent experiments with 3 independent donors. Raw data, or log-transformed data in the case of the NO production assay due to non-normality, were analyzed by one-way analysis of variance (ANOVA) with Tukey’s multiple comparison test or two-way ANOVA in the case of Aβ42 recovery into the circulation of engineered vessels. Data are represented as scatter plots overlaid with calculated means ± 95% confidence interval using error bars. Analysis of Aβ fibrillization in the cell-free thioflavin T assay was performed after fitting to a Boltzman sigmoidal curve using GraphPad Prism 6.05 software then extracting t_50_ and curve maximum from the model.

## 5. Conclusions

A relationship between HDL and brain health has emerged with the discovery of several novel, brain-relevant vasoprotective functions of HDL coupled with epidemiological and animal model data supporting a protective role for HDL particularly with respect to CAA. Evaluation of AD-relevant HDL functions in clinical populations now becomes an important question. To prepare for studies using clinical specimens, many of which are banked and may be of limited volume, it is important to optimize the method by which HDL is prepared, which will determine cost, volume of plasma required, throughput, and the repertoire of the functions that can be studied in vitro. We compared HDL isolated by density gradient ultracentrifugation to apoB-depleted plasma produced by PEG-precipitation in several novel assays of HDL function and found that apoB-depleted plasma is suitable to evaluate some, but not all, of the beneficial functions of HDL that are relevant to AD. Specifically, apoB-depleted plasma can be used in assays of suppressing Aβ accumulation in bioengineered vessels, preventing Aβ fibrillization, and suppressing TNFα-induced vascular inflammation. However, apoB-depleted plasma does not replicate the ability of HDL isolated by ultracentrifugation to suppress Aβ-induced vascular inflammation, improve Aβ vascular clearance, or induce endothelial NO production.

## Figures and Tables

**Figure 1 ijms-20-00462-f001:**
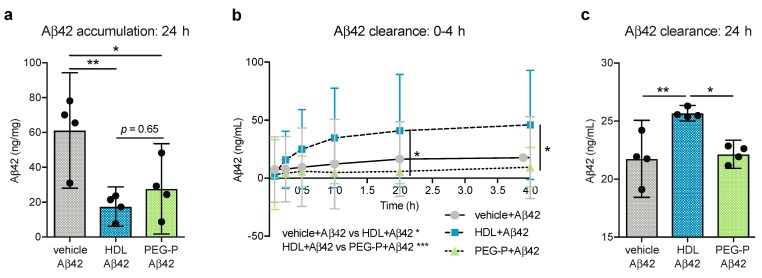
ApoB-depleted plasma reduces Aβ accumulation in bioengineered vessel similar to HDL isolated by ultracentrifugation but does not increase Aβ transport across the vessel wall in the same manner. 3D bioengineered human vessels were subjected to abluminal Aβ42 treatment (1 μmol/L) with or without luminal HDL (200 μg/mL protein, 10.5 mg/dL HDL-C) or apoB-depleted plasma (10.5 mg/dL HDL-C) treatment. Aβ42 level were measured in (**a**) tissue homogenates after 24 h and in circulating media (**b**) over 4 h and (**c**) after 24 h by enzyme-linked immunosorbent assay (ELISA). Scatter plots represent independent experiments with mean ± 95% confidence interval. * *p* < 0.05, ** *p* < 0.01, or exact *p*-value by one-way analysis of variance (ANOVA) with Tukey’s multiple comparisons test for (**a**) and (**c**). * *p* < 0.05, *** *p* < 0.001 omnibus analysis by two-way ANOVA displayed below the graph, * *p* < 0.05 by Sidak’s multiple comparisons test displayed within the graph for (**b**). Aβ42: amyloid beta 42; HDL: high-density lipoproteins isolated by sequential density gradient ultracentrifugation; PEG-P: apoB-depleted plasma by polyethylene glycol precipitation.

**Figure 2 ijms-20-00462-f002:**
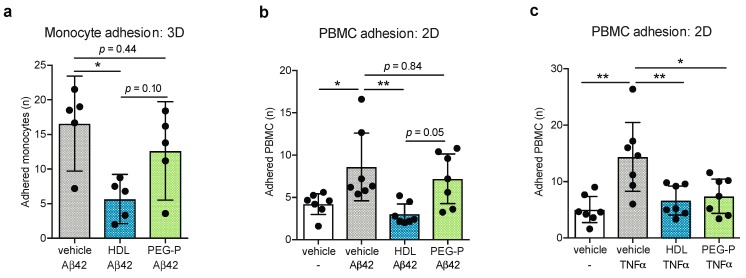
ApoB-depleted plasma does not maintain the ability of HDL to reduce Aβ-induced monocyte binding to the endothelium. 3D bioengineered human vessels were treated with abluminal Aβ42 (1 μmol/L) and luminal HDL (200 μg/mL protein, 10.5 mg/dL HDL-C) or apoB-depleted plasma (10.5 mg/dL HDL-C) for 21 h before fluorescently labelled THP-1 monocytes were added to the circulating media for 3 h. (**a**) The monocytes adhered to the lumen of the bioengineered vessels were visualized with fluorescently microscopy and counted. Brian-derived endothelial cells (hCMEC/d3) in monoculture were pretreated with HDL (100 μg/mL protein, 5 mg/dL HDL-C) or apoB-depleted plasma (5 mg/dL HDL-C) for 2 h then stimulated for 3 h with (**b**) Aβ42 (0.1 μmol/L) or (**c**) TNFα (1 ng/mL) before the addition of fluorescently labelled PBMC for 3 h. The PBMC adhered to the endothelial cells were visualized with fluorescence microscopy and counted. Scatter plots represent independent experiments with mean ± 95% confidence interval. * *p* < 0.05, ** *p* < 0.01, or exact *p*-value by one-way ANOVA with Tukey’s multiple comparisons test. HDL: high-density lipoproteins isolated by sequential density gradient ultracentrifugation; PEG-P: apoB-depleted plasma by polyethylene glycol precipitation; TNFα: tumor necrosis factor alpha; PBMC: peripheral blood mononuclear cell; Aβ42: amyloid beta 42.

**Figure 3 ijms-20-00462-f003:**
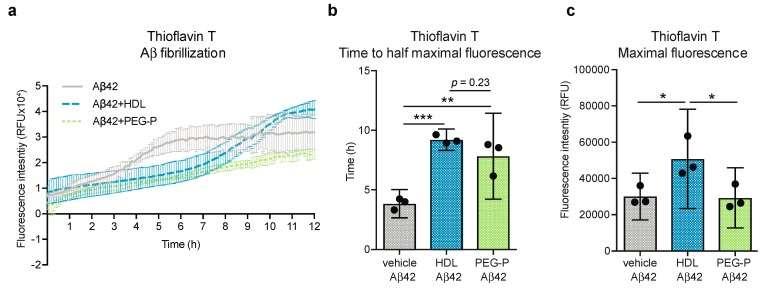
ApoB-depleted plasma is sufficient to assay the anti-fibrillary effects of HDL against Aβ42. (**a**) The formation of β-amyloid pleated sheets was measured in a cell-free assay using thioflavin T over 12 h. Data were fit to a Boltzman curve to determine the (**b**) time to half-maximal fluorescence and (**c**) maximal fluorescence. Scatter plots represent independent experiments with mean ± 95% confidence interval. Time course data is represented with a point for mean ± standard deviation (SD). * *p* < 0.05, ** *p* < 0.01, *** *p* < 0.001, or exact *p*-value by one-way ANOVA with Tukey’s multiple comparisons test. RFU: relative fluorescence units; HDL: high-density lipoproteins isolated by sequential density gradient ultracentrifugation; PEG-P: apoB-depleted plasma by polyethylene glycol precipitation.

**Figure 4 ijms-20-00462-f004:**
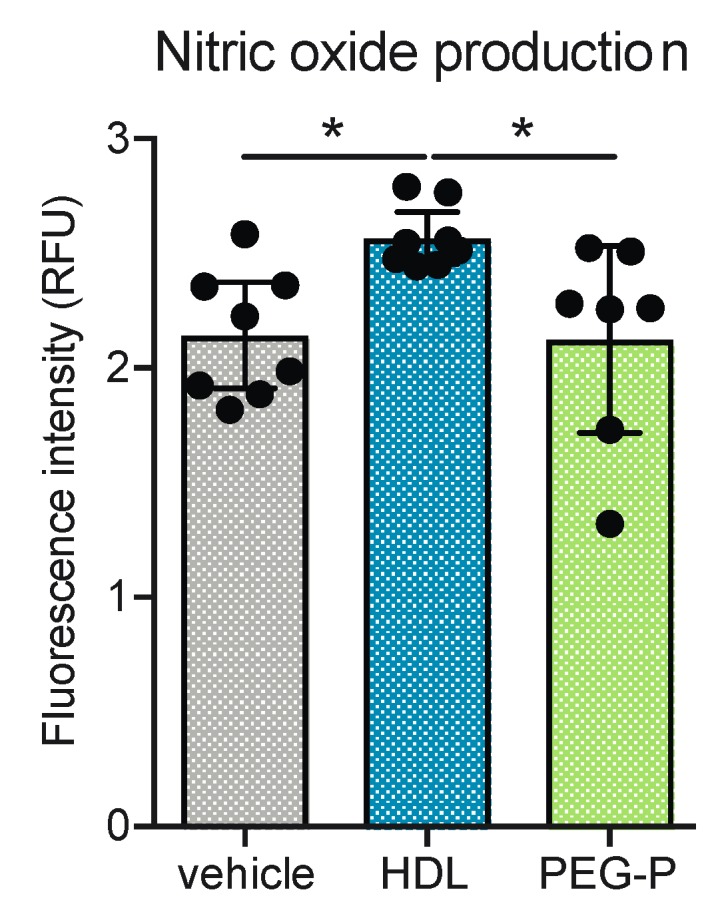
ApoB-depleted plasma does not recapitulate the nitric oxide- (NO) inducing abilities of isolated by ultracentrifugation in human brain-derived endothelial cells. NO production in hCMEC/d3 was measured using the fluorescent indicator DAF-2 DA and HDL treatment (100 μg/mL protein, 5 mg/dL HDL-C) for 4 h. Scatter plots represent independent experiments with mean ± 95% confidence interval. * *p* < 0.05 by one-way ANOVA with Tukey’s multiple comparisons test. RFU: relative fluorescence units; HDL: high-density lipoproteins isolated by sequential density gradient ultracentrifugation; PEG-P: apoB-depleted plasma by polyethylene glycol precipitation; DAF-2 DA: 4,5-diaminofluorescein diacetate.

**Figure 5 ijms-20-00462-f005:**
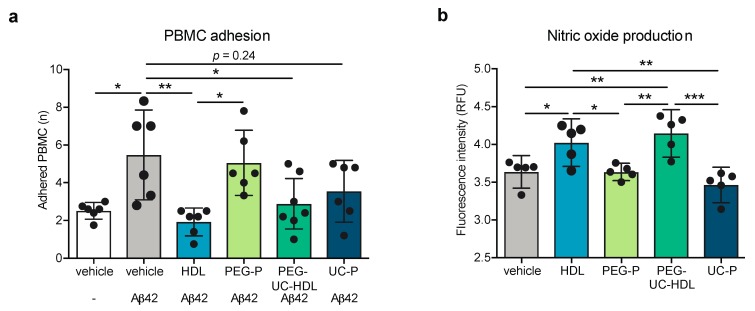
Non-HDL plasma components interfere with some anti-inflammatory activities of HDL is apoB-depleted plasma. (**a**) Adhesion of fluorescently PBMC to hCMEC/d3 was measured using fluorescent microscopy on cells that were pretreated with HDL (100 μg/mL protein, 5 mg/dL HDL-C) or apoB-depleted plasma (5 mg/dL HDL-C) fractions for 2 h then stimulated with Aβ42 (0.1 μmol/L) for 3 h; (**b**) nitric oxide production in unstimulated hCMEC/d3 was measured using the DAF-2 DA and HDL or apoB-depleted plasma fractions treatment for 4 h. Scatter plots represent independent experiments with mean ± 95% confidence interval. * *p* < 0.05, ** *p* < 0.01, *** *p* < 0.001, or exact *p*-value by one-way ANOVA with Tukey’s multiple comparisons test. RFU: relative fluorescence units; HDL: high-density lipoproteins isolated by sequential density gradient ultracentrifugation; PEG-P: apoB-depleted plasma by polyethylene glycol precipitation; PEG-UC: combination method of PEG precipitation and ultracentrifugation HDL; UC-P: 1 step ultracentrifugation HDL; PBMC: peripheral blood mononuclear cell; Aβ42: amyloid beta 42; DAF-2 DA: 4,5-diaminofluorescein diacetate.

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
