# Peer review of "Vasoprotective Functions of High-Density Lipoproteins Relevant to Alzheimer’s Disease Are Partially Conserved in Apolipoprotein B-Depleted Plasma"

_ijms, 2019, doi:10.3390/ijms20030462_

Round 1

Reviewer 1 Report

This Ms deals with an important topic: the apparent vasculoprotective role of HDL components visa the inhibition of Aβ fibres accumulation In central-nervous system cells. The Results are clearly presented and the results discussion is well supported by the experimental results. However, the Ms also deals with methodological issues concerning the maintenance of the protective role of HDL evaluated after isolation of the lipoprotein. Although the HDL  isolation using KBr  differential ultracentrifugation appears to retain the biological properties of HDL,  the authors need to discuss the possibility that the high salt concentrations used and the subsequent de-salting procedures can modify the distribution of HDL-associated apolipoproteins and loosely associated plasma proteins that are responsible for the vasculo-protective properties ( See: Flees et al In Analysis of Fats, Oils and Lipoproteins E. G. Perkins editor American Oil Chemists’ Society 1991: 512-523, and more recent in Ståhlman et al J. Lipid Res 2008; 49: 481-490). Thus it is possible that the differences in HDL biological properties between HDL in apoB-depleted serum and KBr-isolated HDL are  caused by redistribution of HDL-associated proteins differentially affected by the two methods. The authors need to discuss this possibility in the MS. It will be interesting that in future experiments the authors use isolation procedures that have been showing to decrease the dissociation of associated proteins and apolipoproteins of HDL that are caused by the high KBr. The procedure described in the reference by Ståhlman et al,  using Deuterium oxide and Sucrose, that maintains the lipoproteins at physiological salt concentrations, and that is applicable to small plasma sample could be applied to this type of experiments.

Author Response

This Ms deals with an important topic: the apparent vasculoprotective role of HDL components visa the inhibition of Aβ fibres accumulation In central-nervous systemcells. The Results are clearly presented and the results discussion is well supported by the experimental results. However, the Ms also deals with methodological issues concerning the maintenance of the protective role of HDL evaluated after isolation of the lipoprotein. Although the HDL  isolation using KBr  differential ultracentrifugation appears to retain the biological properties of HDL,  the authors need to discuss the possibility that the high salt concentrations used and the subsequent de-salting procedures can modify the distribution of HDL-associated apolipoproteins and loosely associated plasma proteins that are responsible for the vasculo-protectiveproperties ( See: Flees et al In Analysis of Fats, Oils and Lipoproteins E. G. Perkins editor American Oil Chemists’ Society 1991: 512-523, and more recent in Ståhlman et al J. Lipid Res 2008; 49: 481-490). Thus it is possible that the differences in HDL biological properties between HDL in apoB-depleted serum and KBr-isolated HDL are  caused by redistribution of HDL-associated proteins differentially affected by the two methods. The authors need to discuss this possibility in the MS. It will be interesting that in future experiments the authors use isolation procedures that have been showing to decrease the dissociation of associated proteins and apolipoproteins of HDL that are caused by the high KBr. The procedure described in the reference by Ståhlman et al,  using Deuterium oxide and Sucrose, that maintains the lipoproteins at physiological salt concentrations, and that is applicable to small plasma sample could be applied to this type of experiments.

We would like to thank reviewer 1 for this valuable point that is now included in our revised manuscript. As recommended, we addressed the use of high salt solutions and resulting apolipoprotein distribution as a potential drawback of the KBr density gradient ultracentrifugation method that could possibly explain the functional discrepancies observed between the two isolation methods in the discussion section of the manuscript. In each case we referenced the work by Ståhlman et al.

Reviewer 2 Report

The authors have evaluated the potential of functional equivelence of HDL obtained from ApoB-precipitated plasma as compared with HDL isolated by ultracentrifugation in relation to some cerebrovascular functions. Authors have done extensive work on characterizing the impact of the non-lipoprotein component of plasma in AD-relevant assays to evaluate anti-CAA effects by different HDL matrices ex vivo using an outstanding model of 3D bioengineered arteries. The ms is well written, and I would recommend to accept it after minor revision.

General minor comments:

-- > The authors tend to mix previous data to introduce several subsections of results. This is not troubling, but some pieces of text are not strictly data.

-- > In all figures, please check notation of experimental conditions in the x-axis. Some panels of a certain figure (it is the case, for instance, of figure 5) where it is indicate vehicle, HDL, ... with or without Ab42 in the x-axis labels. This difficults the understanding of data.

-- > For instance, the readers may have to assume that Ab42 is infused in panel A as well as it has been in panel B of Figure 2? I know that it is described in the corresponding legend of the figure. The same is found in other figures (Fig. 3, Fig. 4, Fig. 5). The use of different notations among figure panels is confusing. The exposition of cells to Ab42 (final concentration) in each assay should be properly included in each panel figures. May it unified?

-- > The final concentration of HDL should also appear in figure legends in all experiments.

-- > I may also recommend that the final concentration of Ab42 added to the assay (1 µmol/L) should appear in figure legends. How did you reach to check this assay concentration of Ab42 and not other? In this regard, did you check other concentrations (higher) of this Ab and check the protective impact of HDL?

-- > Did the authors check the vehicle condition but non-treated Ab42 (vehicle - Ab42)? It appears in panel A of figure 5, but it is not included in panel B.

-- > All figures, panel annotation for each figure should be (a), (b), ... as indicated in the editorial guidelines of the IJMS journal.

Specific minor comments:

Introduction section

-- > Page 2, lines 71-78: this piece of text might sound better as a starting paragraph or very similar to the summary (see page 8, lines 289-295) in the discussion section. It may be removed from the introduction.

Results section

-- > Page 2, lines 82-83: these two lines are rather introductory and could be inserted in between line 42; particularly, between “...HDL-C levels per se.” and “Mechanistically, preclinical studies...”.

-- > Page 3, lines 101-104: Please provide the circulating leveles of Ab42 into the circulation of bioengeneered vessel; these data are not shown.

-- > Page 3, lines 111-114: this piece of text might be used as a piece of discussion? Strictly results start from line 114 on.

-- > Page 5, lines: in the line of the previous comment: Section 2.4 may rather start from line 158 on.

-- > Page 6, legend Figure 5, insertion of TNFa abbreviation should be removed.

-- > Page 6, Figure 5, please check x-axis notations of panel B (nitric oxide production) (see also general minor comments).

Author Response

The authors have evaluated the potential of functional equivelence of HDL obtained from ApoB-precipitated plasma as compared with HDL isolated by ultracentrifugation in relation to some cerebrovascular functions. Authors have done extensive work on characterizing the impact of the non-lipoprotein component of plasma in AD-relevant assays to evaluate anti-CAA effects by different HDL matrices ex vivo using an outstanding model of 3D bioengineered arteries. The ms is well written, and I would recommend to accept it after minor revision.

General minor comments:

-- > The authors tend to mix previous data to introduce several subsections of results. This is not troubling, but some pieces of text are not strictly data.

We agree that several sections in the Result begin with introductory references to earlier work, which we believe helps to focus the reader toward the specific questions asked in a deliberate manner throughout the Results section. This style is not uncommon and we respectfully request to retain it in the accepted version of the manuscript. Moving this information to the introduction section would necessitate the reader to remember all of the specific background information as they read each section of the Results section and would lead to a introduction section that is less clear and concise. 

-- > In all figures, please check notation of experimental conditions in the x-axis. Some panels of a certain figure (it is the case, for instance, of figure 5) where it is indicate vehicle, HDL, ... with or without Ab42 in the x-axis labels. This difficults the understanding of data.

-- > For instance, the readers may have to assume that Ab42 is infused in panel A as well as it has been in panel B of Figure 2? I know that it is described in the corresponding legend of the figure. The same is found in other figures (Fig. 3, Fig. 4, Fig. 5). The use of different notations among figure panels is confusing. The exposition of cells to Ab42 (final concentration) in each assay should be properly included in each panel figures. May it unified? 

We thank reviewer 1 for this excellent point and have revised the figures to be more clear.

-- > The final concentration of HDL should also appear in figure legends in all experiments.

This information has been added to all figure legends.

-- > I may also recommend that the final concentration of Ab42 added to the assay (1 µmol/L) should appear in figure legends. How did you reach to check this assay concentration of Ab42 and not other? In this regard, did you check other concentrations (higher) of this Ab and check the protective impact of HDL?

This information has been added to all figure legends. The concentration of Aβselected for Aβaccumulation assays, monocyte adhesion assays, and PBMC adhesion assays were determined from dose response experiments in our previous publications investigating these HDL functions (Robert et al 2017, Elife. pii:e29595) (Robert & Button et al2017, Mol Neurodegen, 12(1):60). In those experiments we measured the response of the cells to increasing doses of Aβand selected a dose for future experiments that elicited a significant response compared to vehicle control. References to the dose response experiments in these publications have been added to the methods section of the current manuscript.

-- > Did the authors check the vehicle condition but non-treated Ab42 (vehicle - Ab42)? It appears in panel A of figure 5, but it is not included in panel B.

In panel B, all conditions are without Aβstimulation, this has been clarified in the figure legend.

-- > All figures, panel annotation for each figure should be (a), (b), ... as indicated in the editorial guidelines of the IJMS journal.

Figure panel labels have been changed to comply with the editorial guidelines.

Specific minor comments:

Introduction section

-- > Page 2, lines 71-78: this piece of text might sound better as a starting paragraph or very similar to the summary (see page 8, lines 289-295) in the discussion section. It may be removed from the introduction.

These lines have been removed.

Results section

-- > Page 2, lines 82-83: these two lines are rather introductory and could be inserted in between line 42; particularly, between “...HDL-C levels per se.” and “Mechanistically, preclinical studies...”.

These lines have been moved to the introduction.

-- > Page 3, lines 101-104: Please provide the circulating leveles of Ab42 into the circulation of bioengeneered vessel; these data are not shown.

The specific levels of Aβ42 measured in circulation media from the bioengineered vessels has been added to the text.

-- > Page 3, lines 111-114: this piece of text might be used as a piece of discussion? Strictly results start from line 114 on.

As explained more thoroughly above, we feel that introducing more specific background information immediately before relevant results ensures a better flow of the manuscript rather than adding too many specific pieces of information to the introduction section.

-- > Page 5, lines: in the line of the previous comment: Section 2.4 may rather start from line 158 on.

As explained more thoroughly above, we feel that introducing more specific background information immediately before relevant results ensures a better flow of the manuscript rather than adding too many specific pieces of information to the introduction section.

-- > Page 6, legend Figure 5, insertion of TNFa abbreviation should be removed.

The abbreviation has been removed, we would like to thank reviewer 1 for their close attention to detail.

-- > Page 6, Figure 5, please check x-axis notations of panel B (nitric oxide production) (see also general minor comments).

As noted above, the experiment in panel (b) did not include stimulation with Aβ.  The figure legend has been clarified to reflect this.

Reviewer 3 Report

In the present study the authors examined whether apoB-depleted plasma can be used for evaluation of HDL functions relevant to cerebrovascular health. More specifically, the atuhors used established 3D and 2D in vitro models to assess the capacity of apoB-depleted serum (in comparison to HDL isolated by ultracentrifugation) to delay Aß fibrizillation, induce NO production, as well as inhibit Aß-induced endothelial activation and Aß accumulation within cerebral vessels. HDL outperformed apoB-depleted plasma whose inactivity in some assays most likely reflects interference of albumin and other serum proteins or non-protein components.

Major point:

My major concern is related to NO production. The authors should demonstrate that HDL-induced NO signal is eNOS dependent and for this purpose measure NO production also in the presence of L-NNA or L-NAME.

Minor:

1. In the present Fig. 4 legend the duration of HDL treatment was 4 h and in the Methods 6 h. What is correct?

2. In the Results section, lane 161: '' fluorescence intensity of 2.14 nm to 2.75 nm...'' what is nm?

3. lane 157, the sentence is not OK:..''..in brain is these cells...''

4. Metod used to generate data shown in Fig. 2A is not described in the Methods section.

5. Fig. 2 legend; fluorescently microscopy should be replaced with fluorescence microscopy.

6. Methods, Anti-inflammatory function: Were HDL and apoB-depleted plasma removed before addition of TNFa or Aß and were TNFa and Aß removed before addition of labeled PBMC? or all componentes including HDL/apoB-depleted plasma, TNFa/Aß and PBMC were present in the final incubation? Please clarify this and include in the method description.

Author Response

In the present study the authors examined whether apoB-depleted plasma can be used for evaluation of HDL functions relevant to cerebrovascular health. More specifically, the atuhors used established 3D and 2D in vitro models to assess the capacity of apoB-depleted serum (in comparison to HDL isolated by ultracentrifugation) to delay Aß fibrizillation, induce NO production, as well as inhibit Aß-induced endothelial activation and Aß accumulation within cerebral vessels. HDL outperformed apoB-depleted plasma whose inactivity in some assays most likely reflects interference of albumin and other serum proteins or non-protein components. 

Major point:

My major concern is related to NO production. The authors should demonstrate that HDL-induced NO signal is eNOS dependent and for this purpose measure NO production also in the presence of L-NNA or L-NAME.

We agree that it is important to understand the pathway by which HDL induces NO production and indeed we have previously demonstrated that HDL-induced NO in hCMEC/D3 is eNOS dependent under the same experimental conditions used in the present study (Robert & Button et al2017, Mol Neurodegen, 12(1):60). The text has been revised to reference this previous work (Section 2.4).

Minor:

1.     In the present Fig. 4 legend the duration of HDL treatment was 4 h and in the Methods 6 h. What is correct?

Cells were treated for 4 h, methods section 4.6 has been updated with the correct information.

2.     In the Results section, lane 161: '' fluorescence intensity of 2.14 nm to 2.75 nm...'' what is nm?

The fluorescence intensity units are listed in nanometres erroneously, the correct units should be the arbitrary “relative fluorescence units” and has been adjusted as such in the text and figures.

3.     lane 157, the sentence is not OK:..''..in brain is these cells...''

We have corrected this typo.

4.     Metod used to generate data shown in Fig. 2A is not described in the Methods section.

A method section containing the details to generate these data has been added as section 4.11.

5.     Fig. 2 legend; fluorescently microscopy should be replaced with fluorescencemicroscopy. 

This change has been made.

6.     Methods, Anti-inflammatory function: Were HDL and apoB-depleted plasma removed before addition of TNFa or Aß and were TNFa and Aß removed before addition of labeled PBMC? or all componentes including HDL/apoB-depleted plasma, TNFa/Aß and PBMC were present in the final incubation? Please clarify this and include in the method description.

HDL and apoB-depleted plasma were not removed before the addition of Aβand TNFαand the stimulations were not removed before the addition of labelled PBMC. The methods section has been revised to clarify this point.

Reviewer 4 Report

This manuscript compared HDL and apoB-depleted plasma in terms of inhibiting Aβ vascular accumulation, enhancing Aβ clearance, inhibiting inflammation and promoting endothelial NO production. This information could be interesting to the readers using HDL and apoB-depleted plasma in Alzheimer’s disease and cardiovascular disease research.

1)    Fig. 1B: the x-axis is labeled with ‘Time (min)’. Should it be ‘Time (h)’? A note in the figure says: HDL vs PEGP***, but there is no *** in the figure. The * with vertical line in the figure is confusing.

2)    The title of 2.1 (line 80) and the title of Fig. 1 should be revised. The data in Fig. 1 showed that ApoB-depleted plasma retained the ability of HDL to reduce Aβ accumulation but did not diminish its clearance from the bioengineered vessel.

3)    What is the mechanism for reducing Aβ accumulation but not reducing its clearance from the tissue?

4)    The NO level in the vehicle samples is ~2 in Fig. 4 but is ~3.5 in Fig. 5B. why?

5)    Fig. 5B: PEG-UC should be PEG-UC-HDL.

6)    The abbreviation list in the figure legend of Fig. 5 included TNFα. Was TNFα used in the experiments for Fig. 5?

7)    Line 395-397: move the ‘at 24h’ at line 397 to line 395 behind bioengineered tissues were collected.

Author Response

This manuscript compared HDL and apoB-depleted plasma in terms of inhibiting Aβ vascular accumulation, enhancing Aβ clearance, inhibiting inflammation and promoting endothelial NO production. This information could be interesting to the readers using HDL and apoB-depleted plasma in Alzheimer’s disease and cardiovascular disease research. 

1)     Fig. 1B: the x-axis is labeled with ‘Time (min)’. Should it be ‘Time (h)’? A note in the figure says: HDL vs PEGP***, but there is no *** in the figure. The * with vertical line in the figure is confusing. 

Reviewer 4 is correct and the x-axis label has been changed to “Time (h)”. The statistical comparisons made for Figure 1b has been clarified in the figure legend. In short, the significant statistics displayed below the graph are the omnibus results from a two-way ANOVA comparison taking treatment and time as significant factors. The statistics displayed within the graph are results from Sidak’s multiple comparisons test investigating the specific time points where there was a significant difference between treatment groups. 

2)      The title of 2.1 (line 80) and the title of Fig. 1 should be revised. The data in Fig. 1 showed that ApoB-depleted plasma retained the ability of HDL to reduce Aβ accumulation but did not diminish its clearance from the bioengineered vessel.

We agree with this point and have revised the title of the results section and figure legend.

3)      What is the mechanism for reducing Aβ accumulation but not reducing its clearance from the tissue?

At first glance it does appear puzzling that there can be discrepancies between the Aβaccumulation and Aβclearance assays. We have suggested several possible explanations for this discrepancy in the discussion on page 8, lines 273 to 279.

4)      The NO level in the vehicle samples is ~2 in Fig. 4 but is ~3.5 in Fig. 5B. why?

The differences in relative fluorescence intensity in figure 4 and figure 5b result from a change in the settings on the plate reader used to measure fluorescence in this experiment. Initial experiments in figure 4 used the “optimal” gain setting while experiments comparing for figure 5 were performed at a later date and used the “manual” gain setting on the plate reader. The optimal vs manual gain setting does not appear to alter the fold change results however the absolute relative fluorescence units vary between the two methods. It is typical to use “optimal” gain settings in the early stages of assay development until the best gain can be determined after which it can be set manually.

5)     Fig. 5B: PEG-UC should be PEG-UC-HDL. 

This correction has been made.

6)      The abbreviation list in the figure legend of Fig. 5 included TNFα. Was TNFα used in the experiments for Fig. 5?

The abbreviation has been removed as it was a typo.

7)      Line 395-397: move the ‘at 24h’ at line 397 to line 395 behind bioengineered tissues were collected.

This change has been made.

Round 2

Reviewer 3 Report

Thanks for improving the manuscript. Very nice work!